# Comparative Environmental Assessment of the Iron Fertilisers' Production: Fe-Biochelate versus Fe-EDDHA

Sara Rajabi Hamedani [1], Mariateresa Cardarelli [1], Youssef Rouphael [2], Paolo Bonini [3], Andrea Colantoni [1] and Giuseppe Colla [1,*]

1 Department of Agriculture and Forest Sciences, University of Tuscia, 01100 Viterbo, Italy; sara.rajabi@unitus.it (S.R.H.); tcardare@unitus.it (M.C.)
2 Department of Agricultural Sciences, University of Naples Federico II, 80055 Portici, Italy; youssef.rouphael@unina.it
3 oloBion-OMICS LIFE LAB, 08028 Barcelona, Spain; pb@olobion.ai
* Correspondence: giucolla@unitus.it

**Abstract:** In response to tackling the environmental consequences of fertiliser production, biofertilisers from organic sources are strongly promoted in line with circular economy and maximising resource use. Despite the outstanding potential of bio-based fertilisers for the sustainable development of the agricultural sector, an environmental investigation of these fertilisers is required to replace synthesised fertilisers. Considering the importance of iron as a plant micronutrient and the scientific gap in the environmental assessment of relevant fertilisers, iron-based fertilisers produced in EU and US geographical zones are selected as a case study in this paper. Therefore, this study examines the environmental performance of two iron-based fertilisers (Fe-biochelate and Fe-EDDHA) by the life cycle assessment (LCA) methodology. The LCA model has been implemented in Simapro software by the ecoinvent database and ReCipe 2016 method considering 1 kg iron content as a functional unit. The results revealed that the Fe-biochelate reduced impacts (69–82%) on all relevant categories, including global warming (69%), terrestrial ecotoxicity (82%), and fossil resource scarcity (77%) in comparison with Fe-EDDHA. Soymeal and acetic acid were the main stressors identified in Fe-biochelate production, while phenol, ethylenediamine and glyoxal were the most significant contributors to the impact categories related to Fe-EDDHA. As a result, Fe-biochelate can be considered a more eco-friendly alternative to Fe-EDDHA.

**Keywords:** life cycle assessment; Fe-biochelate; Fe-EDDHA; protein hydrolysate; sustainable fertilization

## 1. Introduction

### 1.1. Fertiliser Production and EU Regulation

Fertiliser production can entail massive environmental impacts regarding the energy demand and raw materials to manufacture it [1]. Natural gas and coal are highly consumed in manufacturing and contribute to climate change. In addition, the raw materials used to produce fertilisers, such as iron oxide minerals and phosphate rock, are mined from natural habitats, leading to environmental destruction and the displacement of wildlife [2].

Therefore, efforts to reduce energy consumption and relevant greenhouse gas emissions of fertiliser production and develop more sustainable fertiliser formulations are essential to mitigate the abovementioned effects.

In this line, the EU fertiliser regulation 2019/1009 [3], also known as the "new EU Fertilizer Regulation", presents the concept of "fertilising products", which includes all substances, mixtures, and microorganisms intended to provide plant nutrients or improve soil fertility. It sets out detailed requirements for the composition, labelling, packaging, and marketing of fertilising products. The regulation also limits heavy metals and other contaminants in fertilisers, ensuring they do not risk human health, animal health, or

the environment. In addition, the regulation encourages the use of organic and waste-based fertilisers, such as compost and biogas digestate. These products can contribute to the circular economy by using waste products as raw materials and reducing reliance on non-renewable resources [4,5]. By integrating circular economy principles and using biofertilisers, farmers can reduce the amount of waste generated and the environmental impact of their operations [6]. They can also improve the resilience of their farming systems by reducing dependence on external inputs. This, in turn, can help mitigate climate change's effects by reducing greenhouse gas emissions, increasing carbon sequestration in soils, and promoting biodiversity. Therefore, fertiliser production from recycling organic waste and recovering wastewater can highlight the significant role of the fertiliser sector in the circular economy.

### 1.2. Plant Biostimulants

Plant biostimulant, as defined by the new EU fertiliser regulation 2019/1009 [3], means a product stimulating plant nutrition processes independently of the product's nutrient content, with the aim of improving one or more of the following characteristics of the plant: nutrient use efficiency, tolerance to abiotic stress, crop quality traits or availability of confined nutrients in the soil and rhizosphere. They are usually applied to plants via the soil or foliar application, and their use is becoming increasingly in demand, especially in horticultural crops. Plant biostimulants have several significant benefits, including ameliorating plant tolerance to abiotic stress, such as drought, enhancing nutrient uptake and utilisation, and promoting plant growth and development [7]. They can also boost soil quality by increasing microbial activity and soil enzyme activities. The use of plant biostimulants is closely related to the concept of circular economy, which aims to reduce waste and promote the efficient use of resources. Plant biostimulants are often produced from organic waste materials rich in bioactive compounds, such as collagen, seaweeds or plant residues; converting these waste materials into valuable plant biostimulants would improve soil health while promoting sustainable agriculture.

An analysis of biostimulant applications over one thousand pairs of open-field data in 180 qualified studies worldwide reported that the add-on yield benefit is, on average, 17.9%, with the highest efficiency in low soil organic matter content and sandy soils [8]. The above findings were often linked to a positive effect of plant biostimulants on plant nutrient uptake and assimilation. Choi et al. [9] observed the yield efficiency of Romaine lettuce and Micro-Tom tomato fertigated with four N levels (2, 5, 10, and 15 mM) and treated by vegetal protein hydrolysate via foliar spray or root drench. The results showed that the addition of protein hydrolysate via root drench effectively increased the N uptake and subsequently increased the lettuce and tomato yield and quality regardless of N levels. The effect of a protein hydrolysate derived from legume seeds on tomato and cucumber growth under Fe deficiency was also evaluated. The results indicated that although the foliar treatments with the protein hydrolysate could not significantly affect growth parameters when plants were grown in full nutrient solution. At the same time, the biostimulant improved the growth performance of Fe-deficient plants [10]. Carillo et al. [11] investigated the effects of foliar application of a legume-derived protein hydrolysate on greenhouse spinach under four nitrogen fertilisation levels (0, 15, 30, or 45 kg N per ha). They reported that protein hydrolysate-treated plants fertigated at 0 and 15 kg nitrogen produced a higher yield than untreated plants by 33.3% and 24.9%, respectively. Furthermore, the sustainability benefits of biostimulants were assessed by Rajabi Hamedani et al. [12]. They estimated the carbon footprint of both root mycorrhisation and foliar applications of vegetal-derived protein hydrolysate for greenhouse-grown zucchini and spinach. They revealed a 7–24% reduction in the global warming potential.

### 1.3. Iron-Based Fertilisers

Iron is a vital micronutrient that plays an imperative role in plant growth, enzyme function and transporting oxygen in every part of a plant—the roots and leaves. Plant iron deficiency disrupts photosynthesis, respiration and chlorophyll production [13]. Iron absorption depends on fertiliser formulation, soil types and root extension [14]. Nevertheless, soil alkaline pH is the most common reason restricting iron uptake. According to [13], 30% of the world's cultivated soils are calcareous, and plants suffer from a shortage of Fe bioavailability; Kraemer et al. [15] reported that Fe concentration in the soil solution with a pH interval between 5.0 and 8.5 ranges from 0.1 to $10^{-3}$ μM, which is 4 to 5 times lower than the Fe concentration required for optimal plant growth [16].

Utilising Fe-based fertilisers would be a systematic strategy to remediate plant deficiency. However, soil application of inorganic Fe salts is inefficient due to its sensitivity to the soil's chemical and physical characteristics. Iron salts can react as Fe oxide and be inaccessible to plants [17]. As a solution, the application of synthetic chelates is recommended. Synthetic chelates bind to metal ions such as micronutrients (e.g., iron, zinc, copper) to form stable complexes [18]. These complexes are used as fertilisers to feed plants with the micronutrients in a form that is readily available and easily absorbed [17]. There are several types of synthetic chelates used in iron fertilisers, including EDTA (ethylene diamine tetraacetic acid), DTPA (diethylenetriaminepentaacetic acid), EDDHA (ethylenediamine-N,N'-bis(2-hydroxyphenylacetic acid), and IDHA (iminodisuccinic acid). These chelates differ in their ability to hold onto iron and their ability to remain stable under different soil conditions [19–21].

Iron chelates can be applied to plants in several ways, including foliar spray, root drench, and soil application. They are commonly used in hydroponic and greenhouse production, where precise nutrient control is essential for optimal plant growth and yield. However, it is important to note that overuse of synthetic chelates can lead to environmental issues, such as nutrient leaching into groundwater and soil acidification from heavy metal accumulation [22,23]. Additionally, the high cost of these fertilisers makes them less accessible to small-scale farmers. As a result, there has been a growing interest in developing sustainable and cost-effective alternatives to synthetic chelates.

In order to overcome the drawbacks of synthetic chelates, biochelates, a type of innovative fertiliser combining vegetal-derived peptides and metal elements, were designed to improve micronutrient availability for plant uptake, leading to better growth and yield [24]. Iron-biochelates offer several advantages over traditional iron fertilisers. First, they are more efficient or at least as efficient as synthetic iron chelates at delivering iron to plants, as the iron-peptide complex is more easily absorbed and utilised [25]. This leads to better plant growth and increased yield. Additionally, Fe-biochelates are less likely to be lost to leaching or runoff since the iron is more tightly bound to the peptides. Another benefit of Fe-biochelates is their environmental sustainability. By improving iron uptake efficiency, less fertiliser is required to achieve the same results. This reduces the fertiliser needed and decreases the risk of pollution from excess nutrients [10,25–28]. Furthermore, biochelates are often derived from natural sources and recycled organic waste, such as vegetal proteins, reducing fertiliser production's environmental impact [29].

While many studies addressed the importance of Fe-chelate fertilisers as effective products on plant nutrition and crop productivity [25,30–32], an environmental analysis of the production process of these fertilisers has remained untouched. Therefore, the current study performs a comparative assertion via a life cycle assessment approach for producing Fe-EDDHA as a synthetic Fe chelate against Fe-biochelate. The research mainly aims to identify environmental benefits and burdens related to the production of two Fe fertilisers different in origins (biobased vs. synthetic). Farmers, scientists, and business leaders following confident approaches to enhance the quality and quantity of crops and policymakers who promote innovative fertilisers for sustainable development are the primary audiences of this study.

## 2. Materials and Methods

The life cycle assessment methodology (LCA) is a holistic approach consisting of four steps: goal and scope, inventory analysis, impact assessment, and interpretation of results. This study assesses and compares the environmental impacts of producing iron fertilisers—Fe-biochelate against Fe-EDDHA. Therefore, the study was conducted in a cradle-to-gate scope considering raw material extraction up to fertiliser production. (Figures 1 and 2). The assessment also covers two geographical zones (Europe and the US). The functional unit selected is 1 kg of the iron content of produced fertilisers.

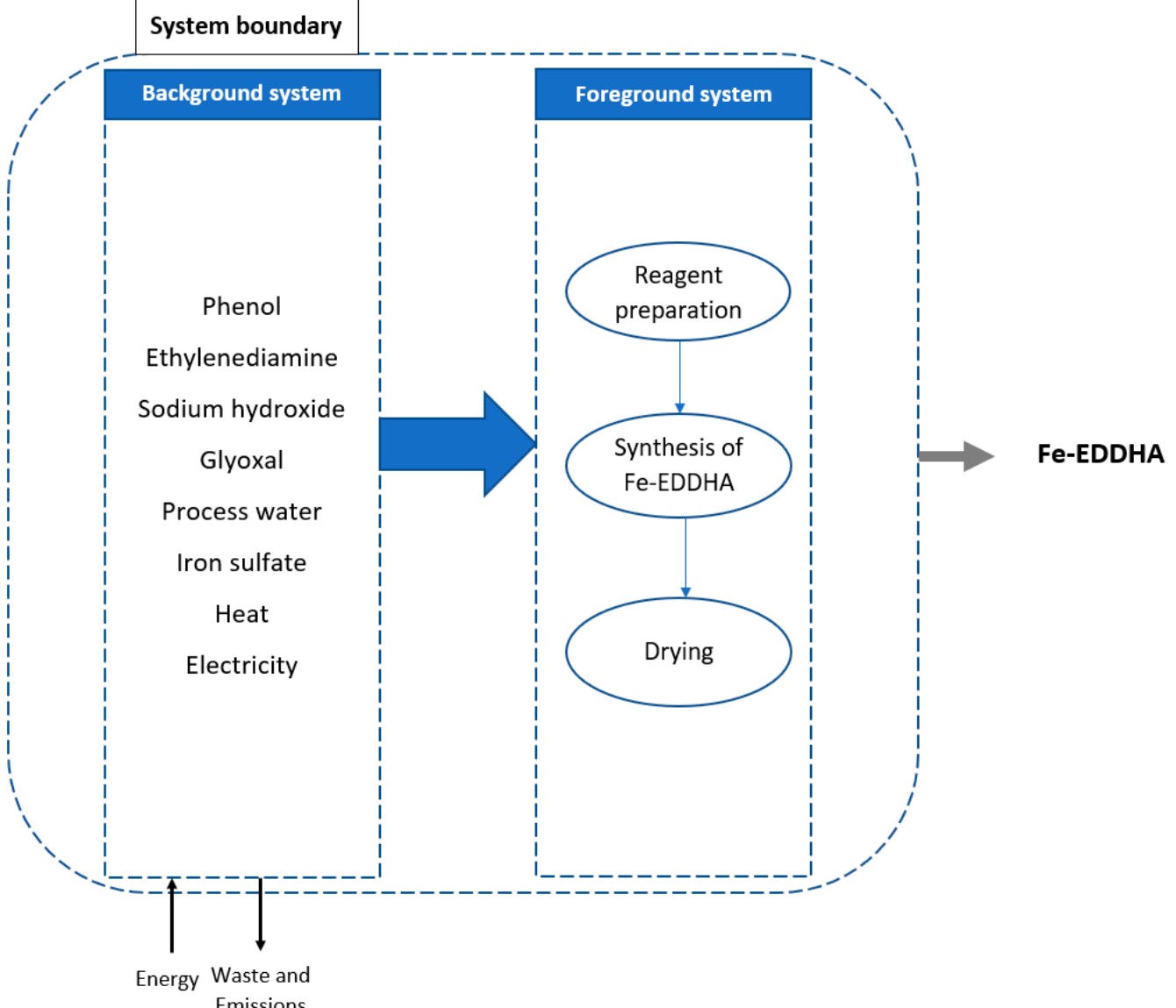

**Figure 1.** The diagram of the system boundary for Fe-EDDHA production.

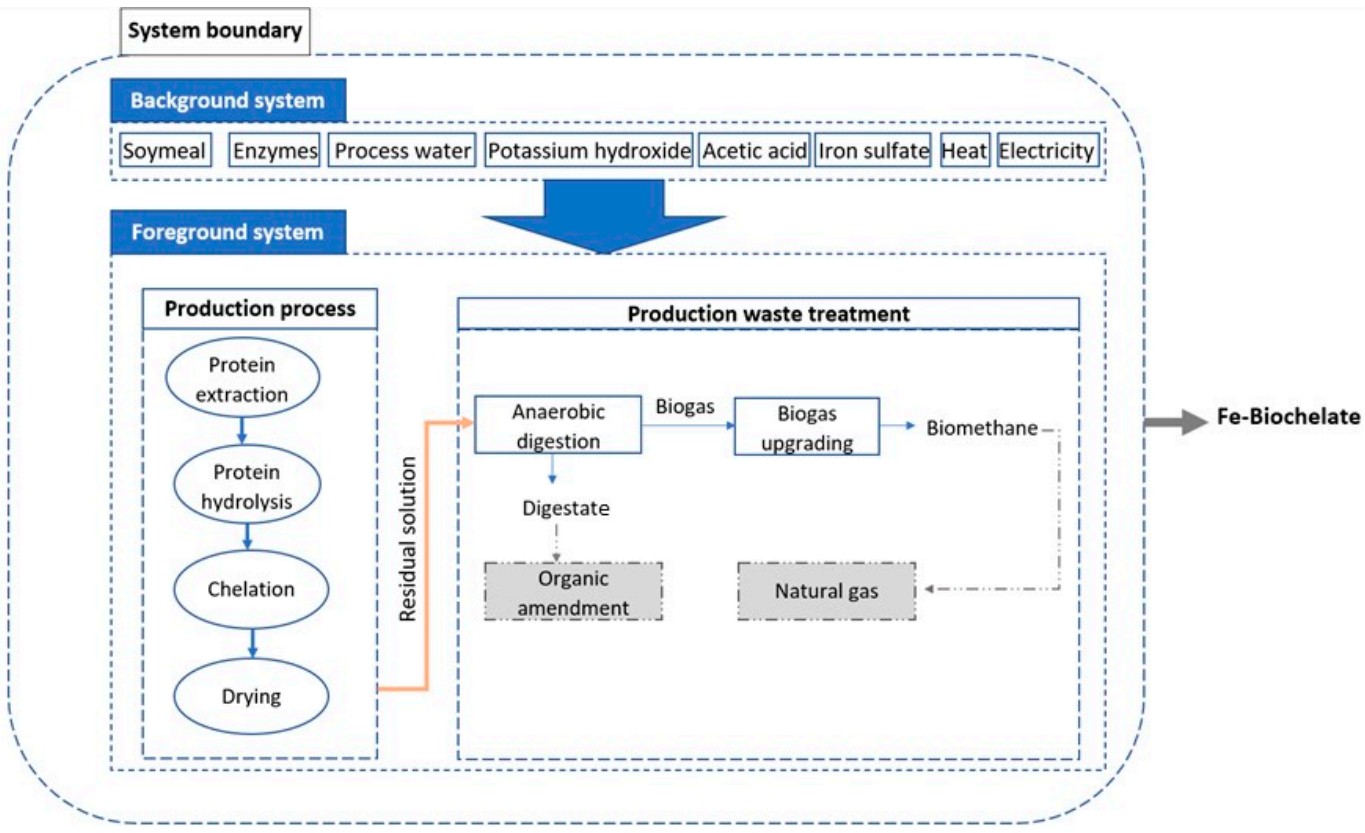

**Figure 2.** The diagram of the system boundary for Fe biochelate production.

*2.1. Life Cycle Inventory*

2.1.1. Fe EDDHA

According to [33], the production technique of the Fe-EDDHA is a single-stage method with outstanding benefits consisting of short process flow, high yield, low production cost, reduced pollution and low energy consumption and no need for an organic solvent. After dissolving phenol in water under 50–100 °C, sodium hydroxide and oxoethanoic acid are dripped into the solution. Under the mentioned temperature, the solution reacts in 2–3 h, and the EDDHA aqueous solution with 40% sodium hydroxide solution is produced. In the next step, dissolved iron ion in water is added to the EDDHA aqueous solution and cooled to 40 °C. The solution needs stirring during 15–30 min to regulate the pH 6.5–7.5. After drying, EDDHA-chelated iron is produced as the ultimate product of this reaction.

Data acquisition related to the production process and all required substances for synthesising Fe-EDDHA were collected based on the previously described patent [33]. The heat and electricity required were estimated considering the different operations such as heating, cooling and drying. Background data associated with the production of energy and substances were taken from the Ecoinvent database V3 [34] (Table 1). Figure 1 reports the steps used for producing Fe-EDDHA as described by [33].

2.1.2. Fe-Biochelate

The foreground data requirements to model the Fe-biochelate product system were provided by the R and D centre of Hello Nature company developing innovative Fe-biochelate fertilisers such as KeyLan Fe. Moreover, background data covering the data to produce input materials (e.g., energy, substances and process water and waste treatment) were extracted from the Ecoinvent database V3 [34] (Table 2). The production waste stream is assumed to be treated by anaerobic digestion; the outputs are biogas, solid and liquid digestate as residual products. Biogas produced after upgrading to biomethane (methane

96% *v/v*) can be substituted for natural gas while digestate is applied due to its nutrient content [34].

**Table 1.** Inventory data for Fe-EDDHA production.

| Items | Unit | Quantity (Unitkg Fe$^{-1}$) |
|---|---|---|
| Output to technosphere | | |
| Fe-EDDHA (6% Fe) | kg | 16.700 |
| Input from technosphere | | |
| *Phase 1*: *reagent preparation* | | |
| Phenol | kg | 6.732 |
| Ethylenediamine | kg | 8.367 |
| Sodium hydroxide | kg | 8.584 |
| Glyoxal | kg | 4.237 |
| Tap water | kg | 12.876 |
| Heat | kWh | 0.224 |
| Electricity | kWh | 0.090 |
| *Phase 2*: *synthesis of Fe-EDDHA* | | |
| Iron sulfate | kg | 5.433 |
| Tap water | kg | 12.876 |
| Cooling | kWh | 0.165 |
| Electricity | kWh | 0.045 |
| *Phase 3*: *drying* | | |
| Heat | kWh | 0.047 |
| Electricity | Wh | 0.770 |

**Table 2.** Inventory data for Fe-biochelate production.

| Item | Unit | Quantity (Unit kg Fe$^{-1}$) |
|---|---|---|
| Output to technosphere | | |
| Fe-biochelate (11% Fe) | kg | 9.090 |
| Input from technosphere | | |
| *Phase 1*: *protein extraction* | | |
| Soymeal | kg | 9.090 |
| Enzymes | g | 6.363 |
| Reclaimed water | L | 54.540 |
| Heat | kWh | 8.444 |
| Electricity | kWh | 2.671 |
| *Phase 2*: *protein hydrolysis* | | |
| Enzymes | g | 18.180 |
| Tap water | kg | 63.630 |
| Potassium hydroxide | g | 454.500 |
| Heat | kWh | 1.863 |
| Electricity | kWh | 1.790 |
| *Phase 3*: *chelation* | | |
| Acetic acid | kg | 3.636 |
| Iron sulfate | kg | 5.000 |
| Heat | kWh | 0.536 |
| Electricity | kWh | 0.090 |
| *Phase 4*: *drying* | | |
| Heat | kWh | 25.270 |
| Electricity | kWh | 3.090 |
| Output to technosphere: waste to treatment | | |
| Residual solution to produce biogas | kg | 5.909 |

*2.2. Impact Assessment*

ReCiPe 2016 v1.03 midpoint method with 18 impact categories converted data inventory related to iron fertilisers—Fe-biochelate and Fe-EDDHA—to conversational indicators. Afterwards, the environmental results were normalised to realise the magnitude and sig-

nificance of the indicators. In the normalisation step, substantial indicators were outstood among others using normalisation factors recommended by [35].

Therefore, those impact categories with immaterial effects have been excluded from the displayed results, and final categories were assessed as follows: marine ecotoxicity, freshwater ecotoxicity, human carcinogenic toxicity, human non-carcinogenic toxicity, terrestrial ecotoxicity, freshwater eutrophication, fossil resource scarcity, global warming, ozone formation-terrestrial ecosystems and ozone formation-human health. The ReCiPe method has been used in other studies on fertiliser production [36–38].

Damage assessment simplifies decision-making based on the value judgment within three environmental impact indicators. Therefore, this study applied the ReCipe endpoint method to evaluate the impacts of iron fertilisers on the damage categories: human health, ecosystems, and resources. Human health damage in DALYs (disability-adjusted life years) represents the years a person has a disability due to diseases or accidents. Ecosystem quality is interpreted by a potentially disappeared fraction of species in terrestrial, freshwater and marine ecosystems during a specific period. The resource scarcity expressed in USD2013 indicates the additional charges for future mineral and fossil resource extraction. The LCA model was developed in SimaPro software (version 9.0.0.49).

## 3. Results

### 3.1. Midpoint Results

Table 3 presents the environmental results regarding the most significant indicators identified for the production processes of Fe-EDDHA and Fe-biochelate. The production of Fe-biochelate can reduce impacts ranging from 69–82% compared to Fe-EDDHA synthesis. Global warming, toxicity and fossil resource scarcity impact decreased by 69%, 75–82% and 77%, respectively. The comparative evaluation of environmental indicators indicated that the effects of all impact categories for Fe-EDDHA production were higher than those of Fe-biochelate production. Therefore, Fe-biochelate was more environmentally compatible than Fe-EDDHA for all impact categories.

**Table 3.** The results of iron fertiliser production processes cradle-to-gate in EU and US scenario (ReCipe, FU: 1 kg Fe).

| Impact Category | Unit | Fe-EDDHA (EU) | Fe-Biochelate (EU) | Fe-EDDHA (US) | Fe-Biochelate (US) |
|---|---|---|---|---|---|
| GWP | kg $CO_2$ eq | 86.32 | 28.64 | 97.03 | 28.59 |
| OF | kg $NO_x$ eq | 0.36 | 0.11 | 0.42 | 0.10 |
| FEu | kg P eq | 0.04 | 0.01 | 0.03 | 0.01 |
| TEc | kg 1,4-DCB | 237.11 | 45.50 | 247.78 | 40.81 |
| FEc | kg 1,4-DCB | 2.37 | 0.67 | 2.38 | 0.57 |
| MEc | kg 1,4-DCB | 3.35 | 0.85 | 3.36 | 0.80 |
| HT | kg 1,4-DCB | 72.44 | 22.30 | 72.83 | 17.56 |
| FRS | kg oil eq | 40.58 | 9.38 | 42.93 | 9.42 |

Global warming (GWP), Ozone formation (OF), Freshwater eutrophication (FEu), Terrestrial ecotoxicity (TEc), Freshwater ecotoxicity (FEc), Marine ecotoxicity (MEc), Human toxicity (HT), Fossil resource scarcity (FRS).

Figures 3 and 4 illustrate the process contribution to each indicator. Regarding geographical zones, Fe-EDDHA produced in Europe entailed lower impacts in all categories than the comparable product in the US. In comparison, Fe-biochelate in Europe had higher burdens on global warming and toxicity-related impacts.

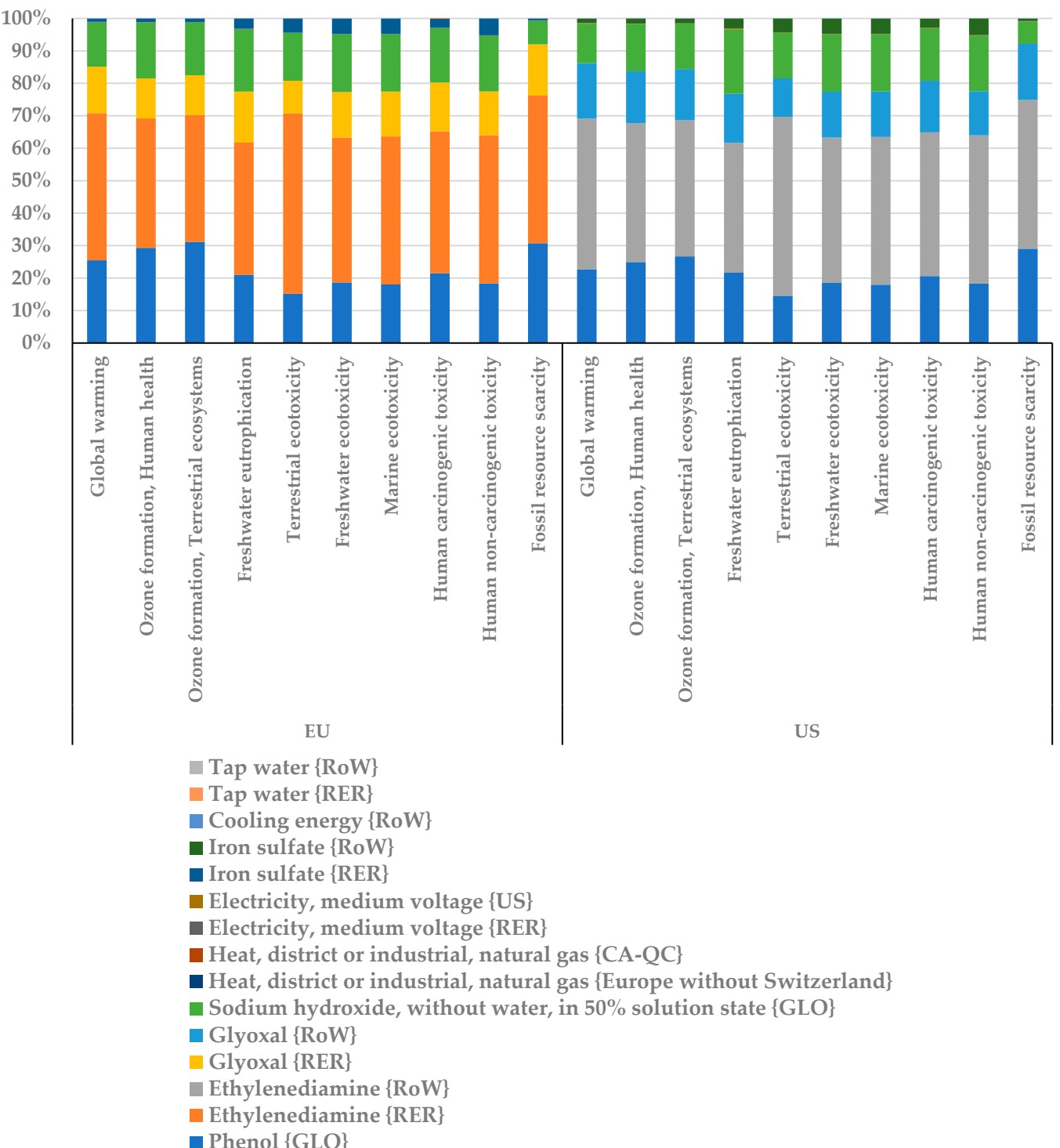

**Figure 3.** Environmental profile and impact categories for producing 1 kg Fe as Fe-EDDHA in EU and US scenario.

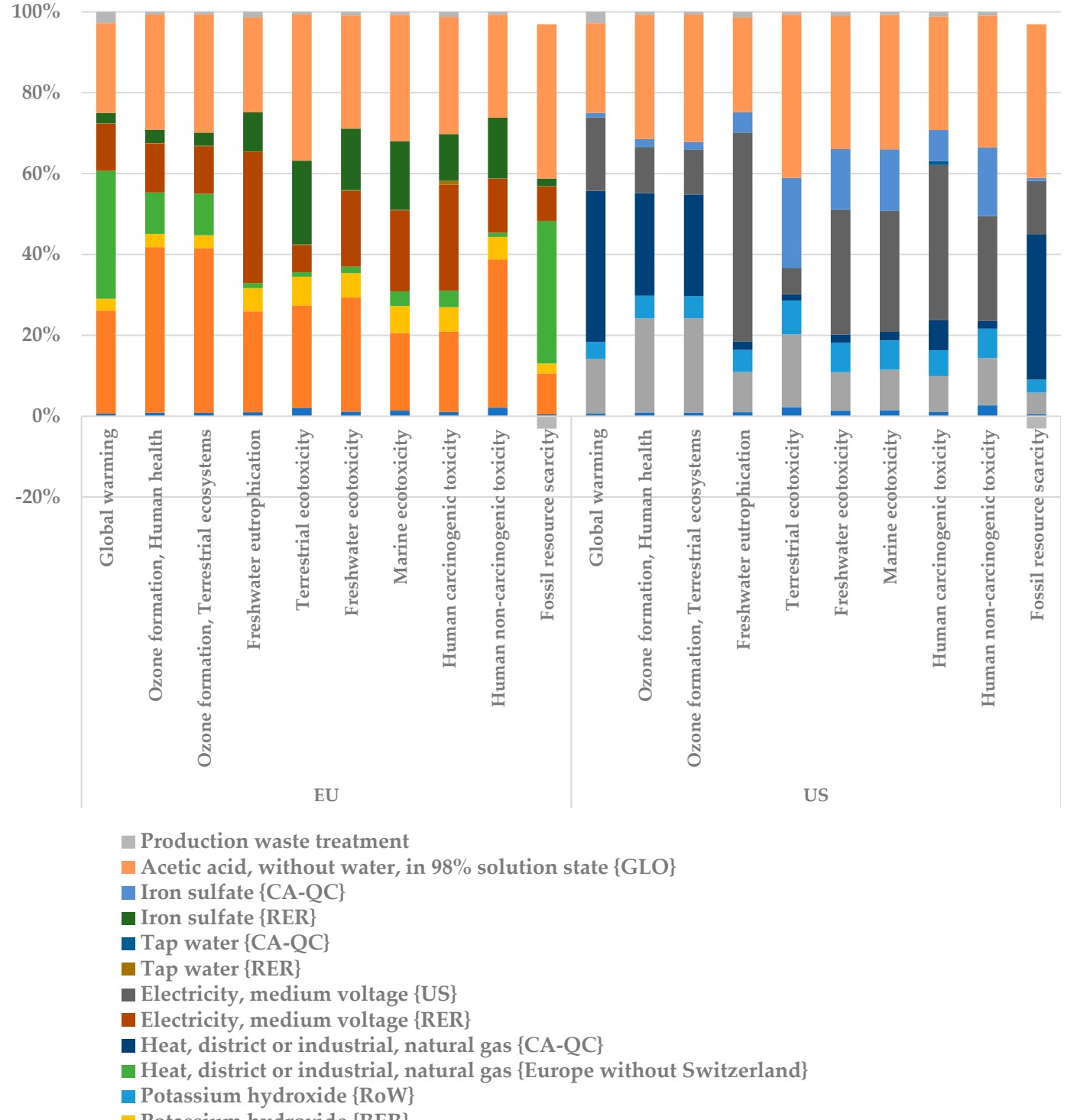

**Figure 4.** Environmental profile and impact categories for producing 1 kg Fe as Fe-biochelate in EU and US scenario.

### 3.1.1. Ecotoxicity

Ecotoxicity, both in terms of aquatic and terrestrial, reflects the effect of hazardous chemical stressors on freshwater. For Fe-EDDHA, the production process of ethylenedi-

amine, followed by sodium hydroxide and phenol, has an enormous impact on terrestrial and aquatic ecosystems (Figure 3).

According to Figure 4, acetic acid, soymeal and iron sulfate, among the other processes involved in Fe-biochelate, had the highest values for terrestrial ecotoxicity-related impacts. The production process of soymeal, acetic acid and electricity had the highest impact on the aquatic extent. Heavy metal emissions to air and water, primarily copper and zinc, arising from the above-mentioned processes were the key contributors to this impact category.

### 3.1.2. Human Toxicity

This category measures the risks of chemicals threatening human health in contact with air and drinking water. Human diseases, including cancer and non-cancer-related risks, are quantified in this category.

The ethylenediamine, phenol and sodium hydroxide production entailed the most significant human toxicity in Fe-EDDHA (Figure 3). Moreover, on the other hand, acetic acid, soymeal, electricity and iron sulfate held the highest values for all the toxicity-related impacts in the Fe-biochelate production process (Figure 4). Airborne copper, nickel and zinc emissions caused cancer and non-cancer-related diseases.

### 3.1.3. Freshwater Eutrophication

This impact category considers the potential for iron fertiliser products to contribute to the nutrient pollution of freshwater ecosystems, which can lead to excessive growth of algae and other aquatic plants resulting in depleting oxygen levels in the water [39].

The primary stressors were ethylenediamine, phenol and sodium hydroxide in the Fe-EDDHA (Figure 3). As illustrated in Figure 4, electricity followed by soymeal and acetic acid, had the highest impact on this category in Fe-biochelate. Phosphate emissions into water were the primary sources of this impact.

### 3.1.4. Fossil Resource Scarcity

This category evaluates the potential impacts of iron fertiliser products on the availability of non-renewable fossil resources. It is quantified by the ratio between the higher heating value of a fossil resource and the energy content of crude oil [40] and measured in kg oil-eq. Natural gas consumed for phenol, ethylenediamine and glyoxal production processes in Fe-EDDHA significantly contributed to the total impact (Figure 3). Acetic acid, soymeal process and heat required for the production phase of Fe-biochelate had the highest impact on fossil scarcity. The natural gas consumption in these processes can justify higher impacts (Figure 4).

### 3.1.5. Global Warming

This category accounts for the greenhouse gases carbon dioxide ($CO_2$), methane ($CH_4$), dinitrogen monoxide ($N_2O$), hydrofluorocarbons (HFCs) and perfluorocarbons (PFCs) in kg $CO_2$ equivalents. Emissions from the production process of phenol and ethylenediamine used in Fe-EDDHA were primarily responsible for global warming (Figure 3). Heat, acetic acid and soymeal had the highest impacts on global warming (Figure 4).

### 3.1.6. Ozone Formation

This impact category allows the assessment of the potential impact of iron fertiliser products on human health related to air quality and the productivity of terrestrial ecosystems, including forests, grasslands and crops in kg $NO_x$ equivalents [41]. Phenol and ethylenediamine significantly contributed to this indicator in Fe-EDDHA (Figure 3). In Fe-biochelate, soymeal and acetic acid were chiefly responsible for this impact (Figure 4).

### *3.2. Damage Assessment*

Table 4 indicates the damage endpoint categories for each iron fertiliser per kg of Fe content. Fe-biochelate reduced impacts on all categories, particularly regarding human

health and resources. In other words, the production of Fe-biochelate results in fewer years that a person is disabled due to a disease or accident and leads to less extra costs in dollars involved for future mineral and fossil resource extraction.

**Table 4.** The endpoint results of iron fertiliser production processes cradle-to-gate in EU and US scenarios (ReCipe, FU: 1 kg Fe).

| Damage Category | Unit | Fe-EDDHA (US) | Fe-EDDHA (EU) | Fe-Biochelate (US) | Fe-Biochelate (EU) |
|---|---|---|---|---|---|
| Human health | DALY | $2.25 \times 10^{-4}$ | $1.93 \times 10^{-4}$ | $6.19 \times 10^{-5}$ | $5.71 \times 10^{-5}$ |
| Ecosystems | species.yr | $4.26 \times 10^{-7}$ | $3.9 \times 10^{-7}$ | $3.54 \times 10^{-7}$ | $3.83 \times 10^{-7}$ |
| Resources | USD2013 | 14.45 | 13.95 | 3.00 | 3.06 |

Disability-adjusted life year (DALY).

In Fe-EDDHA, the ethylene production used in ethylenediamine and glyoxal, followed by benzene manufacturing to synthesise phenol, was mainly responsible for human health and resources damages.

## 4. Discussion

Several LCA studies have been performed on different fertiliser products. Although there is no study that targets iron fertiliser production and the impact assessment results also depend on system boundary, impact assessment method and the functional unit selected, a comparison of published research findings with results obtained from the current iron fertiliser study can be advantageous to understand the generic environmental performance of fertiliser production. Therefore, this section discusses the results of relevant studies without entering numerical comparisons between findings.

Regarding relevant impact categories to fertiliser production, Hasler et al. [38], in an LCA study, analysed the nitrogen and phosphate fertiliser supply chain in Germany. They defined a functional unit as 300 kg of fertiliser applied in one hectare and considered the production, transportation and application of fertilisers in the system boundary. According to the results, the production of fertilisers caused high impacts on climate change, fossil fuel depletion, acidification and resource depletion. Choosing the more efficient raw materials for fertiliser products was recommended to reach a 20% reduction in emissions. Gaidajis and Kakanis [42] identified the impacts related to the production processes of nitrate and compound fertilisers in Northeastern Greece. The results showed that climate change, freshwater eutrophication, and fossil fuel depletion were the most crucial impact categories.

The current study on the production of iron fertilisers also confirms global warming, fossil fuel depletion, and freshwater eutrophication impacts as important categories for fertiliser production.

Regarding the comparative assertion of biofertilisers and conventional fertilisers, Pradel et al. [43,44] compared bio-phosphate fertiliser to mineral phosphate fertiliser (triple super phosphate). They reported that the high energy demand to recover P from sludge led to the environmental superiority of mineral phosphate fertilisers over sludge-based phosphate fertilisers. Indeed, although exploiting bioproducts and waste from other production processes are an appropriate way to minimise waste, products derived from these secondary materials are not always favourable from an environmental perspective. However, in our study, using vegetal-derived peptides for iron chelation was a more sustainable alternative to the synthetic compound EDDHA because of a 69–82% reduction in environmental burdens.

In future studies, the production of different nutrient-biochelate fertilisers (calcium, manganese, zinc and copper) can be analysed and compared against conventional fertilisers. In addition, the system boundary can be expanded to applying biostimulant micronutrient fertilisers in the field (cradle to grave). Furthermore, considering the biostimulant action of biochelates such as vegetal peptides, an increase in root growth and higher C sequestered in the croplands can be expected. In this line, direct and indirect land-use change emissions represented by kg $CO_2$ can be considered in prospective studies.

## 5. Conclusions

Fertilisers have significant environmental impacts, both during manufacturing and in using the fertiliser itself. Environmental assessment of fertiliser production is essential for identifying impacts, evaluating alternatives, informing decision-making and improving transparency and accountability of fertilisers. Therefore, this study has assessed, for the first time in the scientific literature, the environmental profile and impacts associated with the production of two iron fertilisers, Fe-EDDHA and Fe-biochelate, using life cycle assessment in a cradle-to-gate scope. Therefore, this LCA study aimed to identify and compare environmental hotspots derived from energy and material flow in producing two iron-based fertilisers. Innovative Fe-biochelate from vegetal peptides (KeyLan Fe) introduced by Hello Naute company was considered a case study compared with the largely used iron synthetic chelate Fe-EDDHA.

The environmental assessment shows that Fe-biochelate entails much lower environmental effects; soymeal, acetic acid and required heat are the major stressors on impact categories. However, in synthesising Fe-EDDHA as a traditional iron fertiliser, phenol, ethylenediamine, glyoxal, and sodium hydroxide are the main contributor to all environmental burdens. Heat decarbonisation and the application of alternative substances are strongly encouraged within the sustainable development of production systems of fertilisers. Green gas and biomass, heat pumps, waste heat recovery and hydrogen application are a part of the solution. Regarding damage assessment, Fe-EDDHA significantly affects human health and resources compared to Fe-biochelate. According to the overall impacts of the current study, Fe-biochelate such as KeyLan Fe is recognised as the sustainable alternative to Fe-EDDHA. These results provide a generic point of view on optimising the production phase of fertilisers. Because of the disparity in production processes between the types of Fe synthetic chelates, it would be interesting to consider additional Fe synthetic chelate production processes in future LCA studies. Overall, these findings verify that adopting circular economy practices and using biofertilisers can contribute significantly to tackling environmental burdens in agriculture. These practices can help promote sustainable and regenerative, resilient, productive, and environmentally friendly agricultural systems. Future studies can appraise the environmental impacts of the biostimulant micro and macronutrient fertilisers origin from other organic wastes. Furthermore, the economic and social aspects of the waste of fertilisers should be assessed.

**Author Contributions:** Conceptualisation, S.R.H., M.C. and G.C.; methodology, S.R.H. and G.C.; software, S.R.H.; validation, S.R.H., M.C. and G.C.; formal analysis, S.R.H., Y.R., P.B., M.C., A.C. and G.C.; investigation, S.R.H., M.C. and G.C.; resources, G.C.; data curation, S.R.H., M.C. and G.C.; writing—original draft preparation, S.R.H. and G.C.; writing—review and editing, S.R.H., Y.R., P.B., A.C., G.C. and M.C.; visualisation, S.R.H.; supervision, G.C.; project administration, G.C.; funding acquisition, G.C. All authors have read and agreed to the published version of the manuscript.

**Funding:** This work was partially funded by MIUR in the frame of the initiative "Departments of Excellence", Law 232/2016.

**Acknowledgments:** We thank Hélène Reynaud for providing the input data for Fe-biochelate production.

**Conflicts of Interest:** The authors declare no conflict of interest.

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
