# Peer review of "Comparative Environmental Assessment of the Iron Fertilisers’ Production: Fe-Biochelate versus Fe-EDDHA"

_sustainability, doi:10.3390/su15097488_

Round 1

Reviewer 1 Report

Line 66: is not clear "...from 0.1 to 10-3 μM...": do you mean "10.3 μM", or "3 to 10 μM"??

The reference "Goethem TMWJ et al., 2013" is not cited in the text

Line 373: "...Omta S, Olfs H."  is "...Omta SWF, Olfs HW."

Author Response

Line 66: is not clear "...from 0.1 to 10-3 μM...": do you mean "10.3 μM", or "3 to 10 μM"??

We corrected the value in the revised version of the manuscript.

The reference "Goethem TMWJ et al., 2013" is not cited in the text

It was removed in the revised version of the manuscript.

Line 373: "...Omta S, Olfs H."  is "...Omta SWF, Olfs HW."

It has been corrected.

Reviewer 2 Report

The current manuscript focuses may be a pertinent topic and of interest to the journal’s readership, I believe the authors should consider making extensive revisions to the manuscript before it can be considered for publication. 

I was somewhat confused by the language used in several passages as it made me wonder. I’d suggest extensive rewriting throughout for clarity. Perhaps it would be helpful if the authors stated more clearly (in the Abstract and at the end of the Introduction) how the study was conducted; how gather information on ozone, global warming, fossil fuels etc? Any particular database used? Any systematic methodology employed to include/exclude studies? In addition, some sections could probably be written more succinctly  as much of the information seems repetitive.

 I was also confused as to where the figures and tables come from. That is not made clear and appropriate citations and references are not provided. Also, are the data presented in tables mean values? If so, why are measures of variability (e.g., standard deviation) not presented? Or are they the results of statistical analyses? If so, what type of analyses? Neither the captions nor the main text clarify these questions. It is difficult to contextualise what these data mean since no background information is provided as to where they came from and under what conditions they were obtained. In fact, I found that to be a common problem throughout the manuscript. Several results from seemingly previous research are presented throughout the text without any context/background to help readers judge what they actually mean, while several sentences seem to make vague and sweeping statements with no evidence (i.e., reference to existing literature) to back them up.

The Abstract must be improved. It is not clear if the authors are referring to results in general terms or to what they will present in the paper. It does not follow a logical flow and there are some generalised and vague statements (e.g., L19-24). I’d suggest starting with what motivated the research. 

I make similar suggestions for the Introduction but I believe it would be more interesting to start with the broader picture this would be a good opportunity to summarise these challenges to offer readers the ‘big picture’) to then bring in to your topic and how it can help overcome current problems in agriculture). In addition, some passages are confusing. In addition, it is very thin on evidence; several results are presented with no references to existing literature to back them up.

On a similar note, I missed some more insightful concluding remarks; given the results, what are the current gaps in knowledge? Do the authors have any suggestions for the direction of future research? What would be the most pertinent questions to be answered in the coming years?

In addition, I found the manuscript to be very one-sided and provided some other comments in the manuscript file. Finally, the text could benefit from some English language editing throughout for clarity.

Author Response

The current manuscript focuses may be a pertinent topic and of interest to the journal’s readership, I believe the authors should consider making extensive revisions to the manuscript before it can be considered for publication. I was somewhat confused by the language used in several passages as it made me wonder. I’d suggest extensive rewriting throughout for clarity.

Perhaps it would be helpful if the authors stated more clearly (in the Abstract and at the end of the Introduction) how the study was conducted; how gather information on ozone, global warming, fossil fuels etc? Any particular database used? Any systematic methodology employed to include/exclude studies? In addition, some sections could probably be written more succinctly as much of the information seems repetitive.

Answer: Thanks for this comment. Abstract has been modified to clarify the study in an informative way. This study is a life cycle assessment study which must follow the structure of ISO 14040-14044. According to the standards, it consists of four steps goal and scope, life cycle inventory, impact assessment and interpretation. The goal and scope, life cycle inventory and impact assessment have been described in the manuscript in the material and method section as the other life cycle studies and interpretation has been reported in the result section. 

I was also confused as to where the figures and tables come from. That is not made clear and appropriate citations and references are not provided. Also, are the data presented in tables mean values? If so, why are measures of variability (e.g., standard deviation) not presented? Or are they the results of statistical analyses? If so, what type of analyses? Neither the captions nor the main text clarify these questions. It is difficult to contextualise what these data mean since no background information is provided as to where they came from and under what conditions they were obtained. In fact, I found that to be a common problem throughout the manuscript. Several results from seemingly previous research are presented throughout the text without any context/background to help readers judge what they actually mean, while several sentences seem to make vague and sweeping statements with no evidence (i.e., reference to existing literature) to back them up.

Answer: Thanks for this comment. Table 1 reports inputs and output in production process of Fe EDDHA. These data are calculated based on a patent cited in the manuscript [1], In order to clarify the production process, a description has been added to the manuscript. Therefore, data refers to substances required to produce 16.7 kg Fe EDDHA according to the patent (no standard deviation has been considered).

[1]        SICHUAN TONGFENG TECHNOLOGY Co Ltd. Production process for synthetizing EDDHA (Ethylenediamine-N,N’-bis(2-hydroxyphenylacetic acid) ferric-sodium complex) Ferrochel with one-step method, 2010.

Table 2. Reports data of inputs, product (bio-chelate) and waste stream related to production of 9.09 kg bio-chelate. This data is collected from R&D center of Hello Nature company developing innovative fertilizers and they prefer to not disclose process description in details and keep it confidential.

However, production processes of each substances and energy (electricity and heat) required are selected from the datasets in the ecoinvent library called background data in figure 3 and figure 4 the datasets have been addressed. The list of processes extracted from SimaPro for both fertilizers has been added to the Appendix (Table A1, Table A2) and cited in the manucript.

Table 3. presents all environmental results into different indicators. As discribed in “2.2 impact assessment” section, inventory data in table 1 and table 2 by characterization factors of ReCipe 2016 method are converted to these indicators.

The Abstract must be improved. It is not clear if the authors are referring to results in general terms or to what they will present in the paper. It does not follow a logical flow and there are some generalised and vague statements (e.g., L19-24). I’d suggest starting with what motivated the research.

Answer: Thanks for this comment. Abstract has been thoroughly modified. The numerical result have been also inserted to the abstract.

I make similar suggestions for the Introduction but I believe it would be more interesting to start with the broader picture this would be a good opportunity to summarise these challenges to offer readers the ‘big picture’) to then bring in to your topic and how it can help overcome current problems in agriculture). In addition, some passages are confusing. In addition, it is very thin on evidence; several results are presented with no references to existing literature to back them up.

Answer: More references have been added to the manuscript.

On a similar note, I missed some more insightful concluding remarks; given the results, what are the current gaps in knowledge? Do the authors have any suggestions for the direction of future research? What would be the most pertinent questions to be answered in the coming years?

Answer: The conclusions have been modified into providing solutions to reduce environmental impacts of current study and recommendation for future studies.

In addition, I found the manuscript to be very one-sided and provided some other comments in the manuscript file. Finally, the text could benefit from some English language editing throughout for clarity.

Answer: The text has been modified considering your suggestions/comments.

Reviewer 3 Report

The manuscript entitled "Comparative environmental assessment of the iron fertilizers' production: Fe-biochelate versus Fe-EDDHA" was written well and has novel and good results which sounds interesting for the readers. The authors compared two different Fe- based fertilizers: Fe-biochelate and Fe-EDDHA to compare the organic and chemical fertilizers and their effects on different environmental aspects such as global warming, toxicity, fossil resource scarcity, water and air degradation. This is a very good point due to hazardous effects of chemical inputs in environments and finding new alternatives. However, there are some items that should be considered before acceptance for publication, as follows: There are so many studies that were carried out in this field worldwide to find alternative methods and materials rather than chemicals, so, the title is not original and 100% novel but along with other studies have very good and novel results. In my opinion, it can be complementary study with other previous studies but as I mentioned needs to be improved especially in discussion section. It can be as a complementary study with other previous findings. Also, there are few studied focused on effect of fertilizers on global warming as well as fossil resource scarcity and the results of this study can be useful in this manner. There are some typos and grammatical mistakes. It is recommended that the text will be revised by a native English language expert. Discussion part is so short, and it is strictly recommended to be improved. The conclusion was written well. Only the discussion section is very short and it is recommended strictly to be improved. The references are good. I think it should be formatted based on journal format. Also, after revision of the manuscript especially in discussion section, I think there are needed to add more new references in the text as well as at the end of the manuscript.

Author Response

The manuscript entitled "Comparative environmental assessment of the iron fertilizers' production: Fe-biochelate versus Fe-EDDHA" was written well and has novel and good results which sounds interesting for the readers. The authors compared two different Fe- based fertilizers: Fe-biochelate and Fe-EDDHA to compare the organic and chemical fertilizers and their effects on different environmental aspects such as global warming, toxicity, fossil resource scarcity, water and air degradation. This is a very good point due to hazardous effects of chemical inputs in environments and finding new alternatives.

Answer: Dear Reviewer, thank for considering the manuscript written well and with novel and good results which sounds interesting for the reader.

However, there are some items that should be considered before acceptance for publication, as follows: There are so many studies that were carried out in this field worldwide to find alternative methods and materials rather than chemicals, so, the title is not original and 100% novel but along with other studies have very good and novel results. In my opinion, it can be complementary study with other previous studies but as I mentioned needs to be improved especially in discussion section. It can be as a complementary study with other previous findings. Also, there are few studied focused on effect of fertilizers on global warming as well as fossil resource scarcity and the results of this study can be useful in this manner. There are some typos and grammatical mistakes. It is recommended that the text will be revised by a native English language expert. Discussion part is so short, and it is strictly recommended to be improved. The conclusion was written well. Only the discussion section is very short and it is recommended strictly to be improved. The references are good. I think it should be formatted based on journal format. Also, after revision of the manuscript especially in discussion section, I think there are needed to add more new references in the text as well as at the end of the manuscript.

Answer: The discussion part has been modified. There are only a handful life cycle assessment studies on fertilizer production and all of them have been already mentioned in the discussion section. Since the results of life cycle assessment are influenced by methodological aspects such as functional unit, system boundary and impact assessment method and these studies follow different functional unit and system boundary from the current study, the numerical comparison of individual indicators has been excluded from the discussion section and instead results have been compared and discussed from a broader point of view:

  • the most important impact categories reported for fertilizer products;
  • comparative studies of biofertilizer and conventional fertilizers.  

The future research directions have also been added in the discussion section to highlight research gaps. The new references related to introduction section have been added to the revised version of the manuscript.